# Potential Effects of Melatonin and Micronutrients on Mitochondrial Dysfunction during a Cytokine Storm Typical of Oxidative/Inflammatory Diseases

**DOI:** 10.3390/diseases9020030

**Published:** 2021-04-14

**Authors:** Virna Margarita Martín Giménez, Natalia de las Heras, León Ferder, Vicente Lahera, Russel J. Reiter, Walter Manucha

**Affiliations:** 1Instituto de Investigaciones en Ciencias Químicas, Facultad de Ciencias Químicas y Tecnológicas, Universidad Católica de Cuyo, San Juan, Argentina; vmartin@uccuyo.edu.ar; 2Departamento de Fisiología, Facultad de Medicina, Universidad Complutense, 28040 Madrid, Spain; nlashera@ucm.es (N.d.l.H.); vlahera@ucm.es (V.L.); 3Departamento de Fisiología, Facultad de Ciencias Médicas, Universidad Maimónides, Buenos Aires, Argentina; ferder.leon@maimonides.edu; 4Department of Cell Systems and Anatomy, UT Health San Antonio Long School of Medicine, San Antonio, TX 78229, USA; reiter@uthscsa.edu; 5Área de Farmacología, Departamento de Patología, Facultad de Ciencias Médicas, Universidad Nacional de Cuyo, Mendoza, Argentina; 6Instituto de Medicina y Biología Experimental de Cuyo (IMBECU), Consejo Nacional de Investigaciones Científicas y Tecnológicas (CONICET), Mendoza, Argentina

**Keywords:** mitochondrial dysfunction, cytokine storm, melatonin, micronutrients, vitamins, minerals, oxidative-inflammatory disorders, COVID-19

## Abstract

Exaggerated oxidative stress and hyper-inflammation are essential features of oxidative/inflammatory diseases. Simultaneously, both processes may be the cause or consequence of mitochondrial dysfunction, thus establishing a vicious cycle among these three factors. However, several natural substances, including melatonin and micronutrients, may prevent or attenuate mitochondrial damage and may preserve an optimal state of health by managing the general oxidative and inflammatory status. This review aims to describe the crucial role of mitochondria in the development and progression of multiple diseases as well as the close relationship among mitochondrial dysfunction, oxidative stress, and cytokine storm. Likewise, it attempts to summarize the main findings related to the powerful effects of melatonin and some micronutrients (vitamins and minerals), which may be useful (alone or in combination) as therapeutic agents in the treatment of several examples of oxidative/inflammatory pathologies, including sepsis, as well as cardiovascular, renal, neurodegenerative, and metabolic disorders.

## 1. Introduction

Mitochondrial dysfunction is associated with impaired immune and inflammatory responses. It has been suggested that inflammation provoked by mitochondrial dysfunction is responsible for the explosive release of proinflammatory cytokines [1], contributing to multiple oxidative/inflammatory diseases, such as sepsis, neurodegenerative pathologies, inflammatory bowel disease, cardiovascular and metabolic disorders, and respiratory diseases, including complications and death by COVID-19 as one of the most relevant current examples. The systemic hyper-inflammation usually observed in these pathologies is known as cytokine storm or macrophage activation syndrome [2]. 

Notably, the preservation of mitochondrial health through healthy life habits, pharmacological agents, or nutritional supplements provides significantly improved mitochondrial dynamic and activity, reducing inflammation and oxidation stress [1].

In this regard, it is known that melatonin and micronutrients are essential elements that are required by living beings including humans in order to perform multiple metabolic and physiological functions for maintaining an optimal state of health [3,4]. Interestingly, both melatonin and many of these micronutrients exert several of their actions at the mitochondrial level [5].

In this review, we focus on the effects of several micronutrients including vitamin B9 (folic acid), C, D, E, zinc, selenium, magnesium, manganese, copper and, with especial emphasis in the actions of melatonin, on mitochondrial dysfunction, oxidative stress, and cytokine storm observed in some oxidative/inflammatory diseases.

## 2. Meaning, Etiology, Features, and Consequences of “Cytokine Storm”

Cytokine storm is a severe and damaging immune reaction in which there is unlimited and uncontrolled production of pro-inflammatory cytokines. This term has become more generally known in the wake of COVID-19. However, it is not a new concept in medicine [6]. This process also occurs in other situations because of autoimmune conditions such as rheumatoid arthritis [7], infections by bacteria such as *Yersinia pestis* [8], and by other types of viruses [9,10], among other oxidative/inflammatory diseases.

Cytokines are low molecular weight proteins released by various cells that act as messengers between different cells, with auto, para, and endocrine actions. Cytokines are synthesized by different cells, but macrophages and T helper (Th) lymphocytes are the main sources. Their synthesis is usually involved in a cascade of actions, stimulating target cells that produce other cytokines [11]. They are responsible for coordinating an effective immune response according to the infection and regulating the inflammatory process. Cytokines play an important role in the innate and acquired immune response [12] and transmit signals to macrophages and lymphocytes to target the site of inflammation. Cytokines such as interleukins (IL) IL-1α, IL-1β, IL-6, tumor necrosis factor (TNF-α), and monocyte chemotactic protein (MCP-1), among others, contribute significantly to the defenses of normal host. Under physiological conditions, once they have exerted their action and the pathogen has been eliminated, the immune system returns to normal. However, in some situations such as sepsis, the immune system gets out of control. Furthermore, cytokines released into the bloodstream can induce damage to distal organs through the activation of an increasing number of cells of the immune system [13].

The cytokine storm is initiated when large numbers of white blood cells (macrophages, neutrophils, and mast cells) are activated, releasing abundant pro-inflammatory cytokines. The mechanisms of this process depend on phenomena in which an autoinduction of the pro-inflammatory cytokine IL-1 is involved. This cytokine is capable of inducing the expression of its own gene as well as the production of other pro-inflammatory cytokines such as TNF-α and IL-6. In addition, IL-1 induces the gene expression of chemoattractant molecules that mediate the migration of inflammatory cells into tissues. On the other hand, IL-6 is produced by a large number of cells (macrophages, T and B-lymphocytes, endothelial cells…), initiates the activation and differentiation of T cells, and stimulates the differentiation of B lymphocytes as immunoglobulin-producing cells [14]. These processes contribute to IL-6 being an important player in acute inflammation due to its role in regulating the acute phase response, binding to its membrane receptor (mIL-6R) on immune system cells and to its soluble receptor (sIL-6R) [15]. Both interactions lead to stimulate their own expression as well as the massive secretion of various chemotactic molecules [16,17]. This phenomenon generates an amplification cycle that contributes to the recruitment of a greater number of immune cells to the site of infection, as well as the uncontrolled production of cytokines, which characterizes the cytokine storm.

Regarding TNF-α, preclinical studies in mice infected by influenza A virus show how blocking this factor with anti-TNF-α antibodies reduced lung recruitment of inflammatory cells (macrophages and neutrophils). In addition, TNF-α blockade decreased the production of pro-inflammatory cytokines by T cells, lowered the expression of proteins involved in the cascade of the nuclear transcription factor of the kappa light chains of activated B cells (NFkB) signaling pathway, as well as suppressing the severity of the disease without preventing viral shedding. Moreover, blocking TNF-α led to greater control of virus replication by the host [18,19]. NFκB also plays an important role in the initiation of the cytokine storm, because it is capable of inducing the expression of pro-inflammatory genes. This nuclear factor is activated by various mechanisms, such as pro-inflammatory cytokines (IL-1β, TNF-α), a cellular pro-oxidant state, and stimuli related to viral and bacterial antigens [20,21,22].

Concerning COVID-19 disease, whose etiological agent is the severe acute respiratory syndrome—coronavirus 2 (SARS-CoV-2), the cytokine storm could be the main pathogenetic process that leads to the acute respiratory syndrome (ARDS), which is characteristic of the most severe infections [23,24]. The pathophysiological mechanisms underlying the cytokine storm associated with COVID-19 are not fully understood. However, an uncontrolled and generalized inflammatory response that has positive feedback actions has been proposed [25].

During a viral infection, it is important to highlight the involvement of the immune system. Cells of the innate immune system express pattern recognition receptors, including Toll-like receptors (TLRs). These receptors recognize intrinsic molecules that correspond to pathogen-associated molecular patterns (PAMPs), such as lipids, proteins, and nucleic acids of the foreign agent itself [26,27,28]. Regarding SARS-CoV-2 and other coronaviruses, PAMPs are associated with their RNA and are recognized by several TLRs [28,29]. As a result, various signaling pathways and transcription factors such as NFκB are activated. In this process, NFκB translocates to the nucleus to activate the transcription of a large number of pro-inflammatory cytokines. In patients with COVID-19, elevated levels of serum IL-6 are correlated with respiratory failure and ARDS in the most severe cases [30,31]. IL-6 stimulates the expression of C-reactive protein (CRP), which is an acute phase protein used as a serum biomarker for severe infection of this type of coronavirus.

Likewise, the humoral immune response through antibody production and the T-lymphocyte-mediated response plays a very important role during viral infections [28,32]. Data on the humoral immune response to SARS-CoV-2 infection show a specific response of T lymphocytes, both cytotoxic (CD8^+^) and helper (CD4^+^) cells and production of immunoglobulin M (IgM) when the infection is more incipient, and immunoglobulin G (IgG) is produced in later stages.

The responses described derive from both the activation of immune cells and the direct cytotoxic effect of the virus on target cells, resulting in serious organ damage. As a result, the collateral damage caused by the immune response that tries to control the infection may be more dangerous than the pathogen itself.

## 3. Interrelation between Cytokine Storm and Mitochondrial Dysfunction

### 3.1. Mitochondrion: Function, Dynamics, and Dysfunction

Mitochondria are cellular organelles that perform numerous essential functions that ensure homeostasis. They play a crucial role in energy metabolism by synthesizing adenosine triphosphate (ATP) through oxidative phosphorylation. Both glucose and fatty acids metabolites enter the tricarboxylic acid (TCA) cycle to produce ATP [33]. Mitochondria also participate in the regulation of cell calcium homeostasis, apoptosis, and they are an important source of reactive oxygen species (ROS). Mitochondrial ATP generation depends on the transfer of electrons along the transport chain (ETC), which is coupled to proton transport across the inner mitochondrial membrane, establishing an electrochemical gradient that is capable of driving ATP synthesis. Mitochondria are dynamic organelles that change their morphology in response to physiological and pathological stimuli, which also affect their functions. Mitochondrial dynamics comprises several processes: mitochondrial biogenesis, fusion, fission, and mitophagy [34]. A number of pathological situations including cardiovascular, metabolic, and inflammatory diseases are associated with mitochondrial dysfunction. Aberrant mitochondrial physiology appears to be the result of energetics and/or dynamics alterations. Disturbances of mitochondrial ETC not only lead to reduced ATP production but to modifications of mitochondrial morphology [35]. In turn, an imbalance in mitochondrial dynamics decreases the efficiency of mitochondrial energy production [33,36]. Finally, mitochondrial dysfunction is considered the main source of ROS in cells, and it contributes to the development and progression of many pathologies, including inflammation (Figure 1) [37].

### 3.2. Inflammation and Mitochondrial Dysfunction

Inflammation is a complex, protective response of the body to infections and tissue damage. The inflammatory response includes activation of the immune system to repair damaged tissue and defend against pathogens through the secretion of specific mediators. However, when inflammation persists, it produces tissue damage in many diseases.

Many reports have shown the association between the inflammatory process and mitochondrial dysfunction. Inflammation promotes mitochondrial dysfunction, and dysfunctional mitochondrion participates in the pathogenesis of inflammation via several mechanisms, which may establish a vicious circle of recurrent inflammation [38]. The development and progression of several inflammatory disorders are accompanied by mitochondrial dysfunction and enhanced ROS production [39]. Both acute and chronic inflammatory diseases are characterized by the exaggerated generation of oxygen-based reactive species, which produce damage in mitochondrial proteins, lipids, and mitochondrial DNA (mtDNA) (Figure 1). These alterations negatively influence normal mitochondrial function and dynamics [40]. Inducible nitric oxide synthase (iNOS) activity is also elevated in the mitochondria during inflammation, leading to enhanced mitochondrial NO production and reactive nitrogen species (RNS). Both ROS and RNS reduce respiratory chain activity and ATP production, produce mtDNA alterations, and finally lead to cell damage and death [41,42].

#### 3.2.1. Inflammation Alters Mitochondrial Energetics

Inflammatory mediators can also alter activities of the TCA cycle. TNF-α and IL-1 reduce pyruvate dehydrogenase (PDH) activity, together with a concomitant reduction of complex I and II activities [43]. A reduction of PDH activity by inflammatory cytokines has been shown in several cell types such as cardiomyocites, hepatic and skeletal muscle cells, and in cells exposed to septic stimuli [44]. Reduced PDH activity worsens mitochondrial dysfunction, because less acetyl-coenzyme A is produced. Furthermore, this situation reduces the ability of mitochondria to produce melatonin intrinsically, and consequently, it also worsens inflammatory processes [45,46]. Melatonin is a powerful anti-inflammatory agent, and its loss at the mitochondrial level would certainly exaggerate the inflammatory response. Alpha-ketoglutarate dehydrogenase (KGDH) is a rate-limiting enzyme in the TCA cycle, and it is a key element for the mitochondrial ETC [47]. Reduced KGDH activity has been observed during inflammatory conditions including inflamed neural tissue in Alzheimer’s disease [48].

During oxidative phosphorylation (OXPHOS), NADH generated by the TCA cycle is oxidized and provides electrons to the ETC. A reduced expression of genes encoding subunits of complexes I to IV and ATP synthase has been observed in several pathological inflammatory situations. Acute systemic inflammation induced by intravenous lipopolysaccharide (LPS) administration exerts inhibitory effects on OXPHOS [49]. ETC complexes I, II, and IV mRNA and protein levels were down-regulated in muscle and liver cells incubated with LPS [50]. Systemic LPS administration in rodents leads to a TLR4-mediated burst of pro-inflammatory cytokines, which alter mitochondrial energy production. Recent studies showed that TNF-α was able to reduce activity of complex I, III, and IV, leading to decreased ATP production [51]. TNF-α also alters mitochondrial biogenesis regulation in various cell types and tissues through the reduction of peroxisome proliferator-activated receptor-γ coactivator 1α (PGC-1α) expression [52].

#### 3.2.2. Inflammation Alters Mitochondrial Dynamics and Cell Death Pathways

There are many studies of the deleterious effects of inflammatory agents on mitochondrial dynamics. It has been reported that TNF-α induces mitochondrial dysfunction and altered morphology with small condensed mitochondria in adipocytes [53]. This effect may be related to the increased expression of fission protein Fis1 and the reduced expression of fusion protein Opa1. Decreased expression of Opa1 also led to mitochondrial fragmentation [54]. Moreover, IL1-β was able to induce mitochondrial fragmentation and respiration impairment through the fission protein Drp1 in astrocytes. IL-6 is also involved in the lowered expression of peroxisome proliferator-activated receptor-γ coactivator 1α (PGC-1α) and fusion proteins Mfn1 and 2 during the initial stages of cachexia [55]. Finally, other reports also support the interplay between inflammation and mitophagy, leading to apoptosis and cell death (Figure 1) [56].

#### 3.2.3. Mitochondrial Dysfunction Promotes Inflammation

As previously mentioned, inflammation and mitochondrial dysfunction have mutually destructive actions and induce a vicious cycle of function deterioration. Structural and functional alterations of mitochondria can stimulate the production of inflammatory mediators, which in turn can further impair mitochondrial function. It has been shown that ROS are able to promote inflammation through NLRP3 (NOD-, LRR- and pyrin domain-containing 3) inflammasome [57]. Furthermore, mitochondria are considered the main activators of the NLRP3 inflammasome, and they play a crucial role in the control of innate immunity and in the inflammatory response. In fact, interrelationships between mtDNA and NLRP3-inflammasome activation support the involvement of the innate immune pathways in disease coursing with mitochondrial dysfunction [58]. The NLRP3 inflammasome acts as a sensor of mitochondrial dysfunction. Activation of this complex leads to the production of IL1β, which in turn causes loss of mitochondrial membrane potential, reduction of ATP levels, and ROS generation. These findings support the notion that cytokines can produce mitochondrial dysfunction, leading to a vicious degenerative cycle [38]. Another mechanism by which dysfunctional mitochondria evoke inflammatory responses involves the release of damage-associated molecular patterns (DAMPs) into the cytoplasm. DAMPs are macromolecules that are able to induce local inflammatory responses during infections or stress [59]. Due to similarities to bacterial DNA, altered mtDNA can be considered as a DAMP. Moreover, mtDNA is involved in the innate immune response, and thus, specific inflammatory pathologies are directly related to mtDNA alterations [60]. Under normal conditions, defective mitochondrial mtDNA are degraded and eliminated by mitophagy. However, when mitochondrial damage is elevated, mtDNA cannot be effectively eliminated, and it can activate NLRP3 inflammasome-dependent pathways [61].

Activation the TLR-9 pathway is another mechanism by which mtDNA exerts inflammatory responses with the subsequent generation of nitric oxide and TNF-α [62]. It is important to note that normal mitochondria in the presence of TNF-α activate macrophages and cytokine production. Thus, it could be concluded that damaged mitochondria elicit exaggerated inflammatory responses through ROS production and the release of DAMPs, leading to a deleterious vicious cycle (Figure 1).

### 3.3. Relevance of Mitochondrial Dysfunction in the Pathogenesis of Inflammation in Sepsis and COVID-19

Sepsis is a generalized state of inflammation with an initial acute hyperinflammatory phase as a response to infection followed by a hypoinflammatory phase, which is immune-tolerant [63]. Several metabolic processes are re-programmed in each phase, and they play differential roles in the pathophysiological manifestations of the two phases. The hyperinflammatory phase is characterized by increased aerobic glycolysis capacity and oxygen consumption for ETC in many cell types, including monocytes. Activation of the respiratory rate and energy production are essential for the promotion of cytokine production and phagocytosis, which are key elements in host defense mechanisms [64]. This initial phase consists in the reaction of host tissues to enhance energy production in order to increase pathogen-killing capacity of the innate immune cells and attempt to control the spread of the infection [65].

The late hypoinflammatory, immune-tolerant phase is characterized by increased OXPHOS in immune cells [66]. During the second phase of sepsis, mitochondrial respiration and ATP production are partially restored. This hypometabolic state is cytoprotective and immunosuppressive, but it actually impairs recovery and infection control [65]. This phase is associated with a decrease in the cytokine storm and with restoration of sirtuin activity. Sirtuins type 3, 4, and 5 are localized in the mitochondria and are known for their anti-inflammatory and anti-oxidant properties. Increased sirtuin activity promotes OXPHOS and reduces glycolysis [65,66]. Thus, the mitochondrial biogenesis mediators PGC-1α, mitochondrial transcription factor A (TFAM), and nuclear respiratory factor 1 (NRF-1) are upregulated in the second phase of sepsis [67]. As previously mentioned, the release of large amounts of cytokines and chemokines by immune cells produces a sustained systemic inflammatory response that causes the acute respiratory distress syndrome in COVID-19 patients. Indeed, the patients with severe respiratory infection present higher levels of the cytokines than patients with less severe symptoms [68]. Mitochondria seem to be involved in the cytokine storm caused by SARS-CoV-2. A recent analysis of gene expression in the SARS-CoV-2 infected lung cell lines demonstrated an upregulation of genes involved in mitochondrial cytokine signaling and downregulation in the mitochondrial organization, respiration, and autophagy genes [69]. Finally, during the hyperinflammatory phase of COVID-19-related sepsis, immune cells adapt their metabolism, favoring glycolysis over OXPHOS for ATP production. These metabolic changes enhance macrophage phagocytic action and the further synthesis of cytokines and chemokines in a kind of self-perpetuating cycle [70]. Collectively, all these findings provide evidence that mitochondrial dysfunction impairs immune response as well as increases inflammation and severity in the COVID-19-related sepsis. It should be mentioned that OXPHOS and TCA cycle inhibition in mitochondria reduce the synthesis of certain molecules, including melatonin. This supports the use and benefits of melatonin as potential adjuvant treatment strategy to reduce the severity of the COVID-19 [70]. This aspect will be discussed later in this report.

## 4. Role of Melatonin in the Attenuation of Mitochondrial Dysfunction Associated with the Cytokine Storm

The suppression of pineal melatonin as a result of different treatments promotes aerobic glycolysis in immune cells and neutrophil attraction during an inflammatory disease, thereby contributing to the characteristic cytokine storm and mitochondrial dysfunction typical of these pathologies [71,72]. High levels of ATP and the abundant supply of biomolecules produced during aerobic glycolysis support the synthesis and release of the damaging molecules, e.g., cytokines that contribute to the cytokine storm [70]. Conversely, when blood melatonin levels are elevated, it induces the circadian gene Bmal1, which activates the pyruvate dehydrogenase complex, thus promoting oxidative phosphorylation at the mitochondrial level. This is of especial interest in immune cells, where a change occurs in cell phenotype (from reactive to quiescent) when cell metabolism switches from glycolytic to oxidative phosphorylation [71,73,74]. Another means by which exogenous administration of melatonin reverses aerobic glycolysis is by repressing both mTOR and HIF-1α, which disinhibits pyruvate dehydrogenase complex activity and allows the synthesis of acetyl-coenzyme A, which is responsible for mitochondrial melatonin production. Melatonin generated by mitochondria in combination with exogenous melatonin provides a synergistic effect to reduce the intensity of cytokine storm as well as its damaging outcomes [70]. Thus, melatonin regulates immune responses through its mitochondrial effects [71]. In fact, many of the beneficial effects from melatonin administration during inflammatory disease may be due to its actions on mitochondria. In healthy cells, melatonin is synthesized in the mitochondria by aralkylamine N-acetyltransferase/serotonin N-acetyltransferase localized in the mitochondrial matrix. In addition, extra-mitochondrial melatonin accumulates in mitochondria, reaching concentrations higher than those of blood [75]. This probably relates to the active uptake via mitochondrial transporter(s) for this indole. Interestingly, it was observed that conventional antioxidants have a less efficacy than melatonin, since they have a limited access to the mitochondria.

Melatonin protects mitochondria by several actions including inhibition of the mitochondrial permeability transition pore, the elimination of ROS, and the activation of uncoupling proteins. Therefore, melatonin preserves the optimal mitochondrial membrane potential and maintains mitochondrial functions. Moreover, melatonin regulates mitochondrial biogenesis and dynamics, reducing mitochondrial fission and elevating their fusion. Furthermore, melatonin promotes mitophagy and improves mitochondrial homeostasis [76,77,78,79,80,81,82,83].

Sepsis is markedly suppressed by the administration of melatonin, including in humans [84,85]. Moreover, septic mice exhibit a profound inhibition of the respiratory chain activity and a rise in mitochondrial oxidative stress, as determined through the analysis of disulfide/glutathione ratio, lipid peroxidation levels, and glutathione redox cycling enzymes activity. This mitochondrial impairment may be attributed to the increase in the expression of an inducible isoform of mitochondrial nitric oxide synthase enzyme during sepsis, which produces high concentrations of nitric oxide and peroxynitrite at this level. Melatonin treatment counteracts all these alterations, re-establishing redox homeostasis and attenuating the mitochondrial dysfunction observed in this severe systemic inflammatory disease. These melatonin effects are probably mediated by the downregulation of inducible mitochondrial nitric oxide synthase expression and activity during sepsis [42,86,87,88,89,90,91]. In addition, melatonin prevented NFκB activation and cytokine expression, and it preserved glutathione levels, mitochondrial membrane potential, and metabolic activity in endothelial cells cultured under sepsis-like conditions [92]. Furthermore, sepsis triggers alterations linked to complex III in the supercomplexes structure and a decrease in mitochondrial density. Under these conditions, the administration of melatonin to septic mice again prevented mitochondrial dysfunction. Moreover, melatonin improved cytochrome b content and enhanced the assembly of complex III in supercomplexes [93]. Mitochondrial dysfunction is suggested to participate in the inflammatory pathogenesis of septic myocardial depression. A study carried out in a rat septic model showed that melatonin raised the cytochrome c oxidase activity, enhanced heart systolic function, and reduced the mortality rate in these animals [94]. Another report also demonstrated that the treatment with antioxidants, in addition to melatonin, that possess mitochondrial actions including MitoQ and MitoE reduced mitochondrial damage, organ dysfunction, and inflammatory responses in a rat model of acute sepsis, but less effectively than melatonin on some measured parameters [95]. Thus, anti-oxidative actions of melatonin are usually consistent with its anti-inflammatory effects and involve the upregulation of antioxidative enzymes such as superoxide dismutase and downregulation of pro-oxidative enzymes such as nitric oxide synthase, while also directly functioning as a free radical scavenger [96,97], where mitochondrial dysfunction is closely associated with cytokine storm. Thus, ROS play a key role in the pathogenesis of sepsis and liver dysfunction, especially in neonates. In this regard, melatonin had a protective effect on hepatocytes from neonatal rats exposed to H_2_O_2_ (a free radical mediator of septic damage). Collectively, the data document that melatonin preserves the function of many organs during a severe septic infection.

Melatonin may also be useful in the prevention and attenuation of diseases related to aging, which is a major contributing factor in the development of multiple severe diseases such as Alzheimer’s and Parkinson’s disease, cardiovascular pathologies, and others. Aging processes include oxidative stress and damage, mitochondrial dysfunction, and loss of neural regenerative capacity. Aging also involves chronic and acute inflammatory processes, which are often referred to as “inflamm-aging” [98,99,100,101]. Melatonin also has antihypertensive and atheroprotective effects due to its antioxidant and anti-inflammatory actions [102,103,104]. Likewise, several risk factors for cardiovascular disease such as glucose intolerance, hyperinsulinemia, obesity, and dyslipidemia (collectively referred to as metabolic syndrome) are closely related to mitochondrial dysfunction and uncontrolled inflammation. Therefore, melatonin may be useful in the management of this syndrome [105,106,107]. Ischemia–reperfusion injuries are a usual complication in many cardiovascular diseases such as acute myocardial infarctions and stroke or during surgical procedures. A large body of evidence shows that ischemia–reperfusion injuries are mainly caused by an exacerbated generation of ROS. Free radicals consequently trigger an exaggerated inflammatory response, even causing several disabling injuries in distant organs. Related to this, it has been observed that the resistance to ischemia–reperfusion may be dependent on the time of day it occurs, being significantly more severe when happening in the dark-to-light transition (when there is a rapid reduction in melatonin plasma levels). This is because in addition to its powerful antioxidant and anti-inflammatory properties, melatonin is capable of inhibiting the mitochondrial permeability transition pore during reperfusion, which is essential in the attenuation of ischemia–reperfusion injuries [108,109,110]. During traumatic brain injury, melatonin is capable of eliminating damaged or dysfunctional mitochondria by increasing mitophagy through the mTOR pathway (which decreases IL-1β secretion), thus attenuating the exacerbated inflammation observed in this pathology. Hence, melatonin significantly reduces neuronal death and overcomes behavioral deficits after this immunopathological injury [111].

Melatonin reportedly has beneficial effects on physiopathology and the treatment of osteoarthritis, which is a degenerative joint disease characterized by a marked increase in oxidative stress, cytokine storm, mitochondrial dysfunction, and endoplasmic reticulum stress [112]. Likewise, hepatic mitochondrial lipotoxicity (mitochondrial dysfunction caused by exacerbated lipid infiltration of the hepatocytes) observed during the progression of steatohepatitis, accompanied by the consequent increase in oxidative stress and hyper-inflammation observed in this pathology, are also prevented by melatonin [113]. In addition, it has been proved that melatonin significantly raises the Akt/Sirt3 axis activity in an in vitro model of inflammation-induced acute liver failure. This process allows for the maintenance of mitochondrial homeostasis and the restoration of hepatocyte viability in an inflammatory environment associated to mitochondrial dysfunction [114]. The use of melatonin in the treatment of several optic neuropathies and age-related ocular diseases may also be beneficial, since the physiopathology of many of them such as glaucoma, cataract, diabetic retinopathy, and macular degeneration is a consequence, at least in part, of nitro-oxidative stress, hyper-inflammation, and autophagy [115,116,117,118].

As previously mentioned, inflammasomes are intracellular protein complexes that are activated in response to different stress signals and play a key role in triggering the cytokine storm associated to multiple inflammatory diseases. In this sense, chronic obstructive pulmonary disease (COPD) is characterized by a strong prooxidant state and mitochondrial dysfunction, which causes the activation of the NLRP3 inflammasome and the consequent cytokine storm usually observed in this lung pathology. Regarding this, melatonin, administered in both an in vivo and in vitro model of COPD, activated the intracellular antioxidant thioredoxin-1, to inhibit the inflammasome activation and to restore the general antioxidant and anti-inflammatory state by several signaling pathways [119]. As for the prevention and treatment of COPD, melatonin may also be useful in both early stages and in more severe stages of COVID-19 and other respiratory diseases characterized by hyper-inflammation [120,121,122].

The main roles of melatonin in mitochondrion are summarized in Figure 2.

## 5. Contribution of Micronutrients to the Therapy of Oxidative/Inflammatory Diseases through their Mitochondrial Actions

It has been observed that in patients with mitochondrial disease (a set of genetic disorders that leads to a generalized mitochondrial dysfunction), malnutrition as consequence of an inadequate diet may provoke an increase in mitochondrial dysfunction, thus worsening already existing symptoms [123]. It is well known that an adequate nutrition may help maintain optimal immune status, attenuating the impact of infections or other factors that generate a proinflammatory and prooxidant state. Many micronutrients play key roles in strengthening the functions of immune system, thus increasing its resistance against multiple threats. Supplementation with vitamins and minerals is generally a safe and inexpensive means to achieve this objective. However, supplementation should be complementary to a healthy diet and be within suggested safety limits [124,125].

There are several natural therapies based on micronutrient intake (through diet or supplements) that may be used in the treatment of inflammatory/oxidative-based diseases, due to their actions at the mitochondrial level [126]. For instance, B vitamins are crucial in the tricarboxylic acid cycle, whereas selenium and α-tocopherol (vitamin E) are suggested to stimulate the normal function of electron transfer chain. Moreover, selenium participates in mitochondrial biogenesis [5]. Some of the important micronutrients, which make up these therapies, are summarized herein.

### 5.1. Vitamins

When vitamin B9 (folic acid) levels are depressed, it causes mtDNA instability and disorders in this organelle. Thereby, it has been reported that folinic acid supplementation is beneficial in some patients with mitochondrial disease [127,128]. In fact, altered mitochondrial oxidative capacity, fatty acid oxidation, and the mitochondrial biogenesis transcription cascade were significantly improved by folate supplemented diet in mice with cardiac hypertrophy, which is a disease where oxidative stress and inflammation are actively involved [129]. At the cellular level, folate is also involved in the methylation of homocysteine, which becomes methionine: a key intermediary in the generation of glutathione and one of the important intracellular antioxidants. Hence, folic acid can protect from the mitochondrial damage due to oxidative stress through the increase in intrinsic antioxidant concentrations within the cell [130], avoiding the development or progression of oxidative/inflammatory diseases.

Ascorbic acid (vitamin C) is also an important natural antioxidant that prevents lipid peroxidative damage, which is especially observed during vascular dysfunction by acting as scavenger of free radicals, thus re-establishing vascular endothelial functionality. Vascular dysfunction may lead to the development and progression of several oxidative/inflammatory pathologies such as cardiovascular, metabolic, inflammatory, and neurological diseases [131]. In addition, it has been suggested that the administration of vitamin C, selenium, zinc and melatonin, among other micronutrients, are potential “metabolic resuscitators” that are able to recover normal mitochondrial function in the treatment of septic shock, which would improve the survival of the patients with this oxidative/inflammatory disease [132,133,134,135]. Notably, it has been observed that sepsis is usually associated with an acute vitamin C deficiency [136]. Furthermore, results obtained from multiple animal and human studies suggest that there are several therapeutic options targeted to the mitochondria based on the administration of micronutrients for the treatment of mitochondrial dysfunction associated with sepsis. However, neither the optimal dose nor the suitable combination of these micronutrients is yet determined [137].

Vitamin E, both in the form of tocotrienols and tocopherols, has also important antioxidant and antiinflammatory properties, which makes it an attractive micronutrient for improving the treatment and prognosis of aging-related cardiovascular diseases and its associated pathologies [138]. The significant therapeutic potential of vitamin E in the treatment of epileptic seizures associated with neuroinflammation, mitochondrial dysfunction, and lipid peroxidation of multiple brain regions has also been observed. In this regard, it has been determined that vitamin E is a potent cellular antioxidant that acts as a co-factor of several enzymes involved in polyunsaturated lipid metabolism and inflammatory signaling pathways [139]. Additionally, vitamin E in three of its different forms (α-tocopherol, MitoVitE, and Trolox) also was protective against inflammatory, mitochondrial, and oxidative damage in an in vitro model of sepsis [140].

Regarding vitamin D, it has been demonstrated that mitochondria have both receptors for this micronutrient and a functional renin–angiotensin system. The increase in plasma levels of vitamin D inhibits or attenuates the activity of the renin–angiotensin system and improves impaired mitochondrial function during multiple chronic diseases of oxidative/inflammatory etiology such as hypertension, diabetes, neurodegenerative, bone and kidney diseases, and cancer, among others, through its mitochondrial actions [141]. Moreover, one study found that vitamin D prevented neuronal oxidative/inflammatory injuries caused by homocysteine through an increase in the expression of vitamin D receptors at the neural level and the activity of antioxidant enzymes, as well as a reduction in free radical generation [142]. Similar results were obtained in an in vitro model of heart slices with mitochondrial dysfunction induced by homocysteine, again indicating that vitamin D may be useful in the treatment of oxidative/inflammatory-based cardiovascular pathologies [143,144]. Furthermore, vitamin D regulates many age-related processes including autophagy, mitochondrial dysfunction, inflammation, oxidative stress, DNA disorders, and alterations in Ca^2+^ and ROS signaling and epigenetic changes. For this reason, it has been postulated that the aging rate and the development of all age-related pathologies, such as Alzheimer’s and Parkinson’s disease, multiple sclerosis, and others depend, at least in part, on the vitamin D status. Generally, individuals with normal vitamin D plasma levels exhibit slower aging rates than individuals with vitamin D deficiency [145,146]. In addition, vitamin D may attenuate renal injury induced by angiotensin II through the amelioration of mitochondrial dysfunction and modulation of autophagy, suggesting that vitamin D may be useful as a new therapeutic agent for the treatment of chronic kidney disease [147]. Vitamin D deficiency may also be related to an increased risk of COVID-19. This viral disease causes an acute inflammatory process and exaggerated oxidative stress, with severe respiratory consequences, which may lead to death. SARS-CoV-2 infection is associated with mitochondrial dysfunction, oxidative stress, cytokine storm, and apoptosis. Conversely, vitamin D normalizes these altered parameters, reducing even the renin–angiotensin–aldosterone system over-activation observed in this respiratory disease, thus improving the prognosis of the infected patient [148,149,150,151,152,153].

### 5.2. Minerals

Micronutrients such as zinc and selenium are not antioxidant themselves, but they are needed for the activity of several antioxidant enzymes [154]. Particularly, selenium constitutes the active site of at least 25 cellular enzymes called selenoproteins (e.g., glutathione peroxidase, thioredoxin reductase, and methionine sulfoxide reductase), which have important antioxidant and antiinflammatory actions. Indeed, several studies have demonstrated an association between serum selenium levels and immune system status both in humans and in animal models [155]. Selenium dietary intake, mainly which is incorporated into selenoproteins, plays a crucial role in immunity and inflammation. Adequate concentrations of this micronutrient regulate exacerbated immune responses (cytokine storm) and chronic inflammation. Selenium deficiency may negatively affect the activation, differentiation, and proliferation of immune cells. This is mainly associated with increased oxidative stress. Likewise, the incorporation of excessive selenium levels may also impact on immune cell function, affecting many types of inflammation and immunity processes [155,156,157]. Of interest, selenium and selenoproteins have neuroprotective effects during oxidative/inflammatory neurodegenerative disorders. It has been observed that oxidative and inflammatory cell damage induced by glutamate in mouse hippocampal neuronal cell culture was reversed by the overexpression of selenoprotein H in these cells [158]. Similar results were obtained when pretreating mouse cortical neurons exposed to 3-nitropropionic (inducer of oxidative damage) acid with sodium selenite, where the latter protected these cells against free radical damage and stimulated glutathione peroxidase activity [159]. It has also been observed that selenoproteins prevent or suppress the activation of NLRP3 inflammasomes by removing multiple free radicals [160]. Selenepezil, an antioxidant compound based on selenium, showed a higher activity than donepezil in the attenuation of neurocognitive impairment caused by mitochondrial dysfunction, oxidation, and inflammation in Alzheimer’s disease [161]. Another study revealed that plasma zinc and selenium levels were significantly depressed in patients with sepsis [162]. Vitamins C, E, selenium salts, and organoselenium compounds were effective in attenuating oxidative stress and inflammation, both in animal models of sepsis and in septic patients [163,164]. Selenium promotes colon cell mitochondria protection through the upregulated expression of mitochondrial transcription factors, TFAM. Thus, selenium in a high dose could be useful as a potential therapeutic agent in inflammatory bowel disease [165]. It has also been suggested that an adequate selenium intake may inhibit SARS-CoV-2 proteases, this being another beneficial effect of selenium in the prevention and treatment of COVID-19 disease [166,167]. Vitamins C, E, and magnesium have also proven to be useful as adjuvants in the therapy against COVID-19, due to their anti-inflammatory and antioxidant properties [168,169,170,171,172].

For its part, zinc protects cells from oxidative stress by increasing the biosynthesis of glutathione peroxidase and metallothioneins and acting as a cofactor (together with copper) for the superoxide dismutase antioxidant enzyme (CuZn-SOD). Although this enzyme is primarily cytosolic, it is also located in mitochondria and is in charge of metabolizing the superoxide anion, which is a harmful free radical that is abundantly produced during mitochondrial dysfunction observed in oxidative/inflammatory diseases. For this reason, zinc deficiency is usually associated with the increased generation of free radicals and uncontrolled inflammation, as happens in inflammatory bowel disease [173,174]. Similar to melatonin, zinc also attenuates the mitochondrial dysfunction, autophagy, and exacerbated inflammation observed during osteoarthritis, slowing the progression of this oxidative/inflammatory pathology [175]. Zinc also produces anti-inflammatory effects by the inhibition of NFκB signaling pathway and regulation of modulatory T cell functions that may control the cytokine storm observed in COVID-19. Of special interest, zinc status is also closely related to many risk factors for severe COVID-19 such as aging, immune deficiency and metabolic syndrome, which are characterized by zinc deficiency. Accordingly, zinc may act as an adjuvant in the therapy of COVID-19 through reducing inflammation and oxidation, among other factors [176,177,178,179,180,181]. In an animal model of kidney ischemia/reperfusion injury, zinc administration also caused an increase in antioxidant enzyme levels, a reduction in protein and lipid oxidation, and a decrease in pro-inflammatory cytokines levels (IL-1β, MCP-1, and IL-6), attenuating mitochondrial dysfunction and avoiding cell apoptosis [182].

As was mentioned previously, CuZnSOD is located in the mitochondria, which is one of the main target organelles for the prevention and treatment of oxidative/inflammatory diseases associated with mitochondrial dysfunction [183,184]. However, copper is not only required within the mitochondria for the function of SOD but also for the adequate activity of another metalloenzyme, i.e., cytochrome c oxidase, which is the last enzyme in the respiratory electron transport chain. The dysfunction of this metalloenzyme because of low copper levels may also cause free radical overproduction during oxidative/inflammatory diseases [185,186,187]. Copper in free form (unbound from SOD), may also potentiate free radical generation [188,189].

At the mitochondrial level, magnesium, for its part, improves the proton pumps (complexes I, III, and IV) to stimulate extramitochondrial NADH oxidation and to enhance the coupled mitochondrial membrane potential, thus increasing H^+^ -coupled mitochondrial NADPH and ATP synthesis, which is needed for cellular survival [190]. Therefore, magnesium deficiency directly influences the uncontrolled increase in free radical production and in the development of mitochondrial dysfunction, hyper-inflammation processes, and endothelial dysfunction, among other actions [191,192]. Furthermore, magnesium deficiency has a negative impact on the energy generation needed by mitochondria to ATP synthesis and also reduces the general antioxidant capability of cells, which may promote aging due to damage caused by free radical. Both chronic inflammation and oxidative stress were identified as important factors in aging and in several age-related diseases, including metabolic, neurodegenerative, and cardiovascular pathologies, where a marked magnesium deficiency has been observed [193,194,195]. Related to this, in an in vitro model of induced neurodegeneration, it was observed that cell viability correlated significantly with magnesium concentrations. Thus, the treatment with low doses of magnesium resulted in a reduction in ATP production and cell viability with a rise in the levels of ROS and vice versa [196,197]. Similarly, it has been observed that most of the patients with heart failure and type 2 diabetes also have hypomagnesemia. Magnesium supplementation improved cardiac function and insulin resistance in these patients through a decrease in oxidative and inflammatory processes associated with mitochondrial dysfunction [198]. Hypomagnesemia is also a common characteristic in COVID-19 patients. Additionally, magnesium is important in the activation of vitamin D. Therefore, taking into account that magnesium and vitamin D are essentials for adequate immune function, a deficiency in either could contribute to cytokine storm in COVID-19 infection. Consequently, the supplementation with magnesium would be an attractive strategy for the prevention and treatment of COVID-19, especially in populations at risk [199,200].

Similar to copper and zinc, manganese also acts as a cofactor of superoxide dismutase enzyme (MnSOD). Hence, manganese deficiency is directly related to impaired superoxide dismutase function and consequently elevated oxidative stress and hyper-inflammation [201]. In this regard, MnSOD levels are lowered in many oxidative/inflammatory pathologies, including neurodegenerative diseases, cancer, and psoriasis. Conversely, the overexpression of MnSOD in tumor cells may significantly attenuate the malignant phenotype [202]. Similar to copper and selenium, it is well-known that both an excess or a deficiency of manganese and zinc may alter mitochondrial compartments and functions (membrane potentials, oxidative phosphorylation, the tricarboxylic acid cycle, and glutathione metabolism), causing an increase in ROS production and cell death [203,204,205,206,207].

## 6. Conclusions and Prospects

Evidence indicates that oxidative stress and exacerbated inflammation are the foremost standard processes responsible for developing and advancing multiple pathologies, which have become the significant causes of morbidity and mortality in current populations. Therefore, it is of vital importance to direct our efforts to find potential solutions against both events.

Increasingly, mitochondrion appears to be an essential target organelle for antioxidant and anti-inflammatory agents, especially of natural origin. Its complexity and the variety of signaling pathways that occur into this organelle make it possible to attack oxidative/inflammatory diseases from different approaches. The wisdom of nature allows that mitochondria maintain optimal cell homeostasis through their receptors and enzymes that respond to endogenous hormones (e.g., melatonin) and nutritional substances found in natural food (e.g., vitamins and minerals).

Likewise, all extremes are usually wrong; both the insufficient and the excessive uptake of micronutrients or natural supplements may be harmful. For this reason, it is crucial to achieving a balance in our daily habits, thus reaching an optimal state of health according to our features and needs.

As prospects, it would be interesting to investigate the potential synergistic beneficial effects of simultaneous administration of micronutrients and melatonin in patients with oxidative/inflammatory diseases of different degrees of severity (Figure 3).

## Figures and Tables

**Figure 1 diseases-09-00030-f001:**
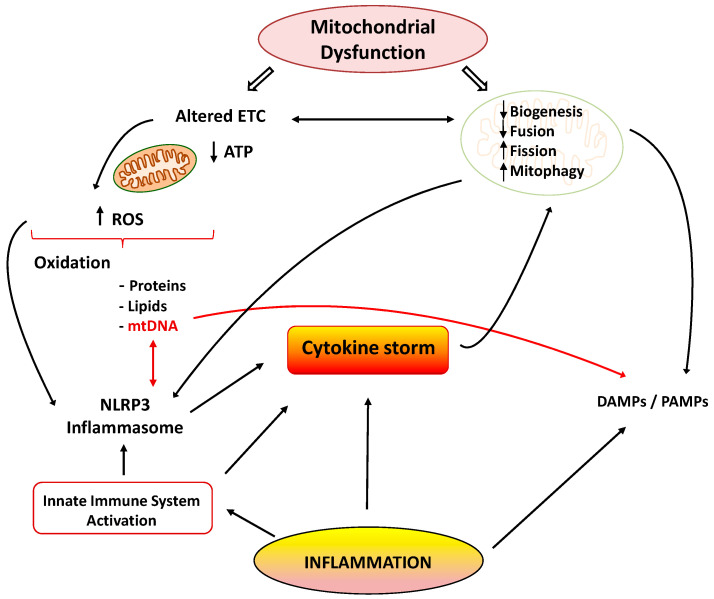
Cytokine storm: link between inflammation and mitochondrial dysfunction. ETC: electrons transport chain; ROS: reactive oxygen species; NLRP3: NOD-, LRR- and pyrin domain-containing 3; DAMPS/PAMPS: damage/pathogen-associated molecular patterns.

**Figure 2 diseases-09-00030-f002:**
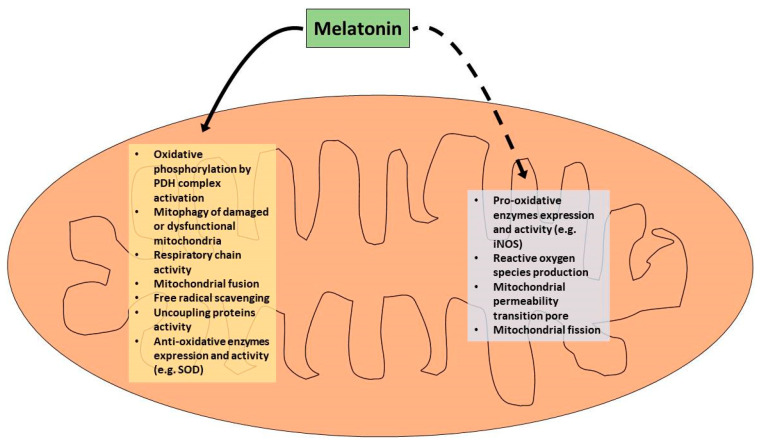
Roles of melatonin in the mitochondrion. The continuous line indicates “stimulation”, and the dashed line indicates “inhibition”.

**Figure 3 diseases-09-00030-f003:**
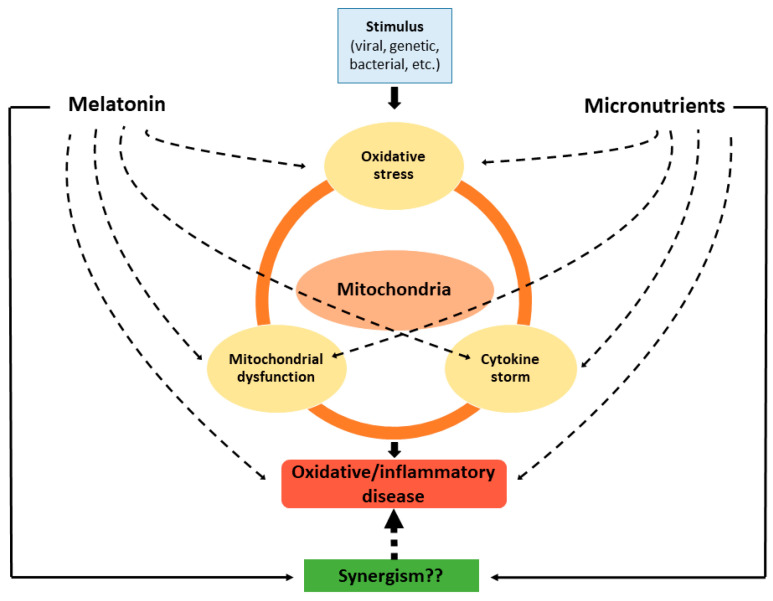
Effects of melatonin and micronutrients on mitochondria in the context of an oxidative/inflammatory disease. Dashed lines indicate “inhibition”.

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
