# Peer review of "Potential Effects of Melatonin and Micronutrients on Mitochondrial Dysfunction during a Cytokine Storm Typical of Oxidative/Inflammatory Diseases"

_diseases, 2021, doi:10.3390/diseases9020030_

Round 1
Reviewer 1 Report
The authors reviewed an interesting area in inflammation topic by collecting the data, including melatonin, vitamins and minerals, about affecting mitochondrial function during inflammation. The review is well written and covers the topic sufficiently to be informative and not to abundant. There are just some minor comments to be considered:
- Lines 74,75 The sentence: "the immune system gets out of control by exercising organic diets." could be ambiguously understood.
- Line 166 Most likely word circle is missing after "vicious".
- Line 279 Instead of term "mitochondrial glycolysis" just "glycolysis" would be more appropriate.
- On several points the commas are missing.
- Since multiple roles of melatonin in preserving mitochondrial function are discussed in the paper, it will be informative to the reader, if these roles of melatonin in mitochondrion are schematically presented in an additional figure.
Author Response
Journal: Diseases
Response to reviewers’ comments
Manuscript ID: diseases-1166487 entitled "Potential effects of melatonin and micronutrients on mitochondrial dysfunction during a cytokine storm typical of oxidative/inflammatory diseases."
Answers to Reviewer #1:
We want to thank the reviewer for his/her comments. We think that the manuscript is now more complete and offers greater clarity in the concepts and positions under consideration.
All changes were marked in red in the text, with and without strikethrough for removing (@) and adding (@), respectively.
Following the reviewer’s suggestion, we changed the sentences in the new version of the manuscripts.
- Lines 74, 75 The sentence: "the immune system gets out of control by exercising organic diets." could be ambiguously understood.
“However, in some situations such as sepsis, the immune system gets out of control."
(Page 2, lines 74-75).
- “Line 166 Most likely word circle is missing after "vicious”.
“Inflammation promotes mitochondrial dysfunction, and dysfunctional mitochondrion participates in the pathogenesis of inflammation via several mechanisms, which may establish a vicious circle of recurrent inflammation [38]”. (Page 4, line 167).
- Line 279 Instead of term "mitochondrial glycolysis" just "glycolysis" would be more appropriate.
“Finally, during the hyperinflammatory phase of COVID-19-related sepsis, immune cells adapt their metabolism favoring glycolysis over OXPHOS for ATP production”. (Page 7, line 280).
- On several points the commas are missing.
The entire text has been revised and some commas have been added (highlighted in yellow) as suggested by the reviewer.
- Since multiple roles of melatonin in preserving mitochondrial function are discussed in the paper, it will be informative to the reader, if these roles of melatonin in mitochondrion are schematically presented in an additional figure.
In complete agreement with the reviewer, we added a new figure (Figure 2) o schematically present the roles of melatonin in mitochondrion mentioned in the text.
Author final note:
We want to thank the reviewer for the careful judgment and acknowledge their contributions. Specifically, the authors believe that the suggestions made have improved the quality of this manuscript and, once more, we would like to express our gratitude for the time invested in this task.
Reviewer 2 Report
The authors have compiled an extensive review of mitochondrial dysfunction as a result of and contributor to inflammation and cytokine storms. The facts used are well referenced and logical. The story is interesting and relevant given the Covid-19 association with ARDS cytokine storms and multiple organ failure. It is difficult in this type of work to keep the organization on track and the authors do a good job with describing overlapping biology that has multiple feed-back and feed forward mechanisms. The choices of what to talk about appear to mostly relate to mitochondrial function and its relation to inflammation and melatonin. The story of minerals and micronutirents, though relevant, is sometimes an awkward fit in the way some paragraphs are written. It may be worth choosing what is relevant to the story by determining who is the star here, the mitochondria or the mediators of mitochondrial health; that may help focus the organization a bit better to tell this highly complex story of so many intersecting paths.
Moderate comments:
- What is meant by "gets out of control by exercising organic diets." in lines 74-and 75? Find another way to clarify that please.
- Some of the early paragraphs on inflammation seem general and under referenced as to mechanistic relevance to mitochondrial health, others are nicely balanced. Seek the balance of depth and facts related to the story you are telling, rather than include facts that are true, but that may not be necessary to this story.
Minor comments:
- A few English verbs could be replaced with better alternatives, for example line 44 "allows to" may read better as "provides".
- "Likewise" in line 126, and "on the other hand" line 102 and other transition words used in conversations can simply be deleted in many places, they do not add value to the scientific writing. Keep them if they add value as true transitions, not just verbal habits.
- In the micronutrient and mineral sections, the vitamin section stays mostly on course, though some paragraphs may deviate into asides. Determine what is valuable to the vitamin portion of this and simply reference the side stories if they become tangents.
Author Response
Journal: Diseases
Response to reviewers’ comments
Manuscript ID: diseases-1166487 entitled "Potential effects of melatonin and micronutrients on mitochondrial dysfunction during a cytokine storm typical of oxidative/inflammatory diseases."
Answers to Reviewer #2:
We want to thank the reviewer for his/her comments. We think that the manuscript is now more complete and offers greater clarity in the concepts and positions under consideration.
All changes were marked in red in the text, with and without strikethrough for removing (@) and adding (@), respectively.
Moderate comments:
- What is meant by "gets out of control by exercising organic diets." in lines 74-and 75? Find another way to clarify that please.
Following the reviewer’s suggestion, we changed the sentence in the new version of the manuscripts.
“However, in some situations such as sepsis, the immune system gets out of control."
(Page 2, lines 74-75).
- Some of the early paragraphs on inflammation seem general and under referenced as to mechanistic relevance to mitochondrial health, others are nicely balanced. Seek the balance of depth and facts related to the story you are telling, rather than include facts that are true, but that may not be necessary to this story.
In the section 3 (Interrelation between cytokine storm and mitochondrial dysfunction), we tried to explain the relationships between inflammation and mitochondria.
Firstly, we briefly explained general aspects on Mitochondrion: function, dynamics and dysfunction (3.1). In the following 3.2 section, we separate in several points the mutual effects of inflammation and alterations of mitochondrial function and dynamics. We considered that this way were easier to understand for the potential reader.
Finally, in 3.3 section, we focused on the Relevance of mitochondrial dysfunction in the pathogenesis of inflammation in sepsis and COVID-19, as a more specific way to explain relationships between inflammation and mitochondrial dysfunction.
We do not clearly understand what the reviewer want to be clarified, arrange or shorten. Thus, we would appreciate a more specific comment in order to rewrite parts of this section.
Minor comments:
Following the reviewer’s suggestion, we changed the sentences in the new version of the manuscripts.
- A few English verbs could be replaced with better alternatives, for example, line 44 "allows to" may read better as "provides".
“Notably, the preservation of mitochondrial health through healthy life habits, pharmacological agents or nutritional supplements provides significantly improved mitochondrial dynamic and activity, reducing inflammation and oxidation stress.” (Page 2, lines 43-45)
- "Likewise" in line 126, and "on the other hand" line 102 and other transition words used in conversations can simply be deleted in many places, they do not add value to the scientific writing. Keep them if they add value as true transitions, not just verbal habits.
We have followed the reviewer's suggestion and both connectors have been removed from the text. (Page 3, lines 103 and 127).
- In the micronutrient and mineral sections, the vitamin section stays mostly on course, though some paragraphs may deviate into asides. Determine what is valuable to the vitamin portion of this and simply reference the side stories if they become tangents.
According to the reviewer’s suggestions, in the micronutrient and mineral sections (especially in the vitamin section), we eliminated some sentences (and their respective references) that may deviate them into asides from the head topic of the manuscript. Moreover, we moved some sentences from each paragraph and joined some paragraphs between them (highlighted in yellow) to improve these sections' fluency.
Author final note:
We want to thank the reviewer for the careful judgment and acknowledge their contributions. Specifically, the authors believe that the suggestions made have improved the quality of this manuscript and, once more, we would like to express our gratitude for the time invested in this task.